# Emotional Imagery Influences the Adaptive Force in Young Women: Unpleasant Imagery Reduces Instantaneously the Muscular Holding Capacity

**DOI:** 10.3390/brainsci12101318

**Published:** 2022-09-29

**Authors:** Laura V. Schaefer, Silas Dech, Lara L. Wolff, Frank N. Bittmann

**Affiliations:** 1Neuromechanics Laboratory, Regulative Physiology and Prevention, Department Sport and Health Sciences, University of Potsdam, 14476 Potsdam, Germany; 2Regulative Physiology and Prevention, Department Sports and Health Sciences, University of Potsdam, 14476 Potsdam, Germany

**Keywords:** Adaptive Force, maximal isometric Adaptive Force, holding capability, neuromuscular adaptation, motor control, pleasant and unpleasant imagery, emotions, emotional imagery, manual muscle test

## Abstract

The link between emotions and motor function has been known for decades but is still not clarified. The Adaptive Force (AF) describes the neuromuscular capability to adapt to increasing forces and was suggested to be especially vulnerable to interfering inputs. This study investigated the influence of pleasant and unpleasant food imagery on the manually assessed AF of elbow and hip flexors objectified by a handheld device in 12 healthy women. The maximal isometric AF was significantly reduced during unpleasant vs. pleasant imagery and baseline (*p* < 0.001, *d_z_* = 0.98–1.61). During unpleasant imagery, muscle lengthening started at 59.00 ± 22.50% of maximal AF, in contrast to baseline and pleasant imagery, during which the isometric position could be maintained mostly during the entire force increase up to ~97.90 ± 5.00% of maximal AF. Healthy participants showed an immediately impaired holding function triggered by unpleasant imagery, presumably related to negative emotions. Hence, AF seems to be suitable to test instantaneously the effect of emotions on motor function. Since musculoskeletal complaints can result from muscular instability, the findings provide insights into the understanding of the causal chain of linked musculoskeletal pain and mental stress. A case example (current stress vs. positive imagery) suggests that the approach presented in this study might have future implications for psychomotor diagnostics and therapeutics.

## 1. Introduction

The interaction between emotions and motor control has been discussed for decades, especially in psychology and behavioral science (e.g., appraisal theory [1,2]) and in psychoneuroimmunology (e.g., mind–body connection [3,4]). It is known that emotions influence different body systems, such as the autonomic, endocrine and motor systems [5]. The link between emotions and motor control can be explained by the central areas involved in processing both emotions and motor control; e.g., the cerebellum, the basal ganglia and the cingulate cortex [5,6,7,8,9,10]. This was discussed in detail in Schaefer et al. [11]. There is broad consensus that mental health issues and complaints of the musculoskeletal system are connected [12,13,14,15,16,17,18,19,20,21]. However, the detailed causal relationship between, e.g., mental stress and musculoskeletal pain, is still unknown [12,18,19,21]. Investigations regarding the influence of mental stress on muscular activity mostly have been performed by evaluating electromyography (EMG) [22,23,24]. Muscular activity is usually higher during stress, e.g., for lumbar and thoracic muscles, while being exposed to negative emotional pictures and music vs. positive ones [25], or during anger and sadness recall interviews vs. baseline values [12] and for the trapezius muscle, during increased self-reported stress induced by the Stroop color word test and mental arithmetic tasks [26], or while anticipating a nociceptive stimulus (uncontrollable and unpredictable) [27]. This leads to the conclusion that mental stress can increase muscle tension. However, there is also evidence that mental stress can cause muscle weakness, which is referred to as psychogenic or functional weakness [28,29,30,31]. Its etiology still remains unclear [29]. Therefore, it would be of interest investigating the effect of mental stress or emotions on muscle strength. There is scarce scientific literature on that particular topic. A significant strength increase was found after an 8-week intervention of weight lifting in a group with prior induced positive emotions vs. controls in elderly people, suggesting that training effects are higher by inducing positive emotions [32]. Mehta and Agnew investigated, inter alia, muscle endurance and force-related changes at 15%, 35% and 55% of the maximal voluntary isometric contraction (MVIC) during a mental arithmetic task [33]. Since force-related changes were not present, the authors concluded that the “performance measure employed may not be sensitive to capture force-related changes” [33]. However, muscle endurance was significantly reduced during the mental stress task [33]. This is in line with another study in which a reduced time to task failure at 20% of the MVIC was reported during a mental-math task [34]. Those studies point out that there are changes in the motor output, especially during sustained isometric muscle action. It was suggested to differentiate two types of isometric muscle actions: the holding (HIMA) and the pushing one (PIMA) [35,36]. During HIMA, time to task failure was significantly reduced compared to PIMA [35,36,37,38,39,40]. This led to the assumption that HIMA is characterized by more complex control strategies than PIMA [11,35,36,41]. Therefore, the investigation of the holding capacity might be an interesting approach to examine the effects of emotions and mental stress.

The Adaptive Force (AF) is based on HIMA, whereby the holding activity is challenged in particular due to the required adaptation to an increasing external load. This must require even more complex neuromuscular control processes than isometric actions without adaptation to varying external forces. By executing AF, the muscular length-tension control must work properly to maintain stability (isometric position) during the external force increase. If the muscle starts to lengthen during this force rise, the maximal holding capacity (maximal isometric AF; AFiso_max_) is exceeded, but the force increases further during the eccentric phase. The maximal AF of the trial refers then to the maximal eccentric AF. Hence, the maximal AF (AF_max_) can be achieved either during isometric or eccentric muscle action. In case of a stable adaptation, AFiso_max_ is similar or considerably high related to AF_max_ or to MVIC, respectively [11,41,42]. In case of instability (inadequate adaptation), muscle lengthening starts at a low force level (decreased AFiso_max_), whereby AF_max_ reaches a similarly high level as for stable adaptations but then during eccentric motion [11,41]. That suggests the holding capacity has to be clearly differentiated to other force parameters. Due to the complex control processes that are assumed to be necessary for a stable adaptation in the sense of AF [11,41,43], it was proposed that the maximal holding capacity is especially sensitive to inputs entering the involved complex control circuitries. Therefore, its investigation might be more beneficial than, e.g., examining the (pushing) MVIC. In first studies, the AFiso_max_ was found to be significantly reduced in healthy participants by perceiving unpleasant vs. pleasant odors [41] as well as by imagining unpleasant vs. pleasant food experiences [11], which was an immediate effect. In perceiving the positive stimuli, the holding capacity switched instantaneously to stability. Both unpleasant odors and unpleasant food imagery are related to the emotion ‘disgust’. Perceiving those negative stimuli, the participants were no longer able to adapt their muscle tension by maintaining the muscle length (isometric conditions) appropriately during the force increase. Moreover, slight mechanical oscillations seem to play a relevant role for stable adaptations between two interacting persons [36,44,45]. Oscillations occur in the form of minimal mutual swinging motions at least of both involved extremities which are in contact. They usually emerge under stable conditions with a frequency around 10 Hz [36,44,45]. In previous studies, the force signal showed an onset of oscillations (AFosc) in the course of force increase on a significantly lower level during pleasant stimuli (imagery/odors), compared to unpleasant ones [11,41]. Furthermore, during stable adaptation (positive odors/imagery) the oscillations appeared still under isometric conditions, whereby for unstable adaptation (negative odors/imagery) they appeared—if at all—after the maximal isometric holding capacity was exceeded, thus, during muscle lengthening [11,41].

The primary findings of those exploratory studies led to the assumption that muscular stability during adaptation to external forces might be impaired by perceptions (imagery/odors) related to the emotion disgust. Since the previous study concerning the effect of emotional imagery on AF had some methodological limitations (no blinding, no randomization, no baseline AF), the aim of the present study was to verify those findings by investigating the influence of pleasant and unpleasant emotional imagery on AF in healthy individuals in a revised and improved design including randomization of imagery tasks, single-blinding (double-blinding is not possible, see Methods) and baseline measurements.

The following main hypotheses were adopted from the previous study: (1) The AFiso_max_ is significantly reduced during unpleasant food imagery compared to during pleasant imagery and baseline. (2) The maximal AF (AF_max_) shows no significant difference between baseline, pleasant and unpleasant imagery. (3) Oscillations occur on a significantly higher AF level (AFosc) during unpleasant vs. pleasant imagery and baseline, but AFosc during pleasant imagery vs. baseline shows no significant difference. In addition to the group comparisons, a case example of a participant being in an actual stressful situation will be presented. The effect of positive imagery under those circumstances will be exemplified thereby.

If the previous findings are positively verified, they might show that the holding capability could be a suitable biomarker to test the immediate effect of emotions on motor output, which is relevant to different fields such as neuromuscular control, neuroscience, psychology, sports and movement sciences and medicine. It might provide, furthermore, a better understanding of pathomechanisms of musculoskeletal complaints related to mental health issues, as will be discussed. A positive verification might result in innovative diagnostic approaches for mental but also particular physical health states.

## 2. Materials and Methods

The study was performed in the Neuromechanics Laboratory of the University of Potsdam (Potsdam, Germany). The AF of elbow and hip flexors of one side was investigated in healthy females by the manual muscle test (MMT) executed by one experienced female tester (35 years, 168 cm, 55 kg; 8 years. of MMT experience) using a handheld device to gauge reactions to pleasant and unpleasant food imagery which provoked a feeling of ‘disgust’ or ‘pleasure’. Food imagery was chosen since it is an easy, harmless and ethically justifiable way to trigger either unpleasant or pleasant emotions under comparable circumstances. It was previously shown that experienced testers (including the abovementioned female tester) are able to perform the MMT reproducibly [43], which is a prerequisite for the present study. In addition to the tester, two assistants participated and conducted the measurements: the first assistant was responsible for handling software, controlling the handheld device, and protocolling; the second assistant guided the imagery task.

### 2.1. Participants

A priori power analysis (G*Power 3.1.9.7) for group differences (dependent *t*-test, two-tailed) on the base of the parameter with the lowest effect size of the previous pilot study ($\frac{\mathrm{AFosc}}{{\mathrm{AF}}_{\max}}$, *d_z_* = 1.390) [11] revealed a necessary sample size of at least *n* = 9 (α = 0.05, β = 0.95). In the anticipation of possible dropouts and due to an assumed lower effect size because of the improved design with a presumably lower bias, *n* = 12 participants were measured.

A total of 12 healthy young women volunteered to participate in the study (age: 24.92 ± 3.50 years; body mass: 64.08 ± 7.69 kg; body height: 170.67 ± 7.63 cm) (for detailed information, see Appendix A). Inclusion criteria were female sex (to generate a homogeneous group), age between 18 and 35 years and good overall health (values > 0 on mood and physical well-being numeric analogue scales from −4 (worst) to +4 (best)). On the measurement day, the participants reported their mood on that scale as 2.58 ± 0.79 and their physical well-being as 3.00 ± 1.22. Exclusion criteria were current or previous (last six months) diseases or health complaints, current feeling of stress and an ongoing or planned psychological treatment. Furthermore, an impaired neuromuscular function of the tested muscles assessed by the MMT prior to the measurements led to exclusion.

The study was conducted according to the guidelines of the Declaration of Helsinki and was approved by the Ethics Committee of the University of Potsdam, Germany (protocol code 35/2018; 17 October 2018).

### 2.2. Handheld Device for Recording Adaptive Force

The reliable and valid handheld device (Figure 1a; development funded by the Federal Ministry of Economy Affairs and Energy; project no. ZF4526901TS7) had already been utilized in previous studies [11,41,43]. It consists of strain gauges (co. Sourcing map, model: a14071900ux0076, precision: 1.0 ± 0.1%, sensitivity: 0.3 mV/V) and kinematic sensor technology (Bosch BNO055, 9-axis absolute orientation sensor, sensitivity: ±1%) and records the reaction force, the accelerations and angular velocity (gyrometry) between tester and participant during the MMT. The sampling rate was 180 Hz. The data were buffered, AD converted and sent by Bluetooth 5.0 to a tablet. An app (Sticky notes, comp.: StatConsult, Magdeburg, Germany) saved the transmitted data [11,41].

### 2.3. Manual Muscle Testing

The MMT is a clinical method testing AF as a biomarker of neuromuscular functioning [46]. The so-called “break test” [43,46] was used in the present investigation, which is usually conducted in submaximal intensities. The aim of the MMT is not to break the participant’s muscle or to test the maximal strength of the participant, but their neuromuscular capability to adapt to the external force increase. In general, the tester applies a gradually increasing force during the MMT by pushing against the limb of participant, who should resist this force application in an isometric holding manner. In case of an optimal adaptation during force increase, the muscle length stays quasi-isometric during the entire MMT until an oscillating force equilibrium is reached between tester and participant on a considerably high force level [11,41,43,46]. If the adaptation is not optimal, the muscle starts to lengthen during the force increase (breaking point) on a considerably low force. The subjective rating of the MMT by the tester is differentiated into two qualities [11,41,43,46]: ‘stable’—the participant’s limb maintains the isometric position during the entire force increase; and ‘unstable’—the participant`s limb gives way, thus, the muscle starts to lengthen and merges into eccentric muscle action during the force increase. The MMT and its interpretation are originally subjective due to the manual assessment. An objectification can be achieved by utilizing the abovementioned handheld device, which records the dynamics and kinematics simultaneously during the test. The characteristics of the force profile applied by the tester are shown in Figure 1b and were described in detail previously [11,41,43]. The tester’s force increase should allow the participant to adapt to the load. For that, firstly, tester and participant come into contact at a low force level for ~1 s; secondly, the tester starts to increase the force smoothly in an exponential way. A smooth force rise is necessary at the beginning, so that the neuromuscular system of the participant has the chance to adapt to the rising force (for neurophysiological explanations see [43]). Thirdly, a linear force increase follows. If an oscillating force equilibrium between tester and participant is reached, this should be sustained for a few seconds, whereby the maximal AF (AF_max_) is attained. The interaction is stopped by the tester and the force decreases again. The duration of the entire force increase (phase 2 to 4) should be ~4 s (Figure 1b). A reproducible force application is a necessary precondition for valid data. As mentioned above, experienced testers are able to perform reliable force profiles over time [11,41,43]. The tester of this study proved her ability to test reproducibly prior to the study by performing 10 repeated force increases against a stable resistance [43].

### 2.4. Questionnaires

Two questionnaires had to be filled in, the first one online prior to the measurement day. It examined anthropometric data, current or planned psychological counselling, sensory perception of food consumption while eating (e.g., odor, taste, optics, texture) as well as the three most tasty (pleasant) and the three most disgusting (unpleasant) foods. The foods had to be rated on a scale from −4 (most unpleasant) to +4 (most pleasant). Those were the base for food imagery during the MMT and were discussed prior to the measurements to obtain precise instructions for the imagery tasks (see below).

The second questionnaire was filled in on-site. The current state of mood and physical well-being were obtained (see above). General health questionnaires, such as the SCL-90 (HTS), were not applied since the investigation did not aim at psychological well-being. Coming from movement sciences, the abovementioned scale seems to be sufficient to obtain an impression of the self-reported mood and well-being for the purposes of this investigation.

### 2.5. Food Imagery

The queried food preferences were discussed between participant, tester and assistants on the measurement day. The participant described the food experience in as much detail as possible and the tester noted the exact words used. Then, the tester asked for individual memorable experiences of the two most pleasant and two most unpleasant foods, e.g., smells, tastes, associations, etc. The aim was to identify triggering and exact words used by the participant to ensure a well-executed imagery task during the MMT trials. Furthermore, the participant chose the most suitable picture of the respective food on a tablet (pictures were from a common online search engine) provoking pleasure or disgust. Across all participants, the most pleasant foods included brownies, pasta with salmon, strawberries, chocolate, pancakes, mango, ice cream, pizza, curry with rice, sushi, BLT sandwich, lasagne, potatoes with quark, mustard eggs, yeast dumplings or hamburger; the most disgusting foods ranged from meat, fish, blood sausage, beetroot, octopus, offal, bananas, Brussels sprouts, spinach, tomatoes on a sandwich, aspic and licorice to pickled gherkins. This highlights the very individual food choices perceived as pleasant or unpleasant. Overall, the unpleasant foods were rated in 13 cases with −4, in 10 cases with −3 and in one case with −2; the pleasant foods were rated 19 times with +4 and five times with +3. In all cases, the foods were connected to individual experiences. For example, the participant who named “BLT sandwiches” as one of her most pleasant foods (+4) said they reminded her of the positive situation of “being on family vacation in Sweden, sitting in the winter garden and watching the lake” while “eating a BLT sandwich with fresh toast, mayonnaise, lettuce, flavorsome bacon and tomatoes”. She reported this as an “experience of great pleasure”. The same participant chose aspic as her most unpleasant food (−4). She described it as “glibbery with pieces of meat inside, it wobbles in the mouth” and “you don’t know what is inside”. This example underlines the highly individual experiences connected with the food consumptions including the related positive or negative emotions.

### 2.6. Setting of Manual Muscle Testing

The starting positions of the MMTs of elbow and hip flexors including the handheld device are illustrated in Figure 1 (according to [11,41]). For testing the elbow flexors, the participant lay supine on a practitioner table and flexed her elbow joint to 90° with a maximal supination (Figure 1c). The tester had the handheld device in her palm and contacted therewith the distal forearm of the participant. To avoid a probably painful pressure on the forearm, the handheld device was cushioned. For the starting position of the hip flexor test, the participant also lay supine with a hip and knee angle of 90° (Figure 1d). The tester contacted the distal end of the participant’s thigh with the handheld device. Cushioning was therefore unnecessary. To ensure reproducibility, the placement of the device was marked at the respective limb. The force vector of the test was in the direction of muscle lengthening of the participant’s elbow flexors (elbow extension) or hip flexors (hip extension). Visual contact between tester and participant was prevented by a curtain on a hanging rail (Figure 1c,d). During the MMTs, the participants were asked to hold their starting position stable (isometrically) for as long as possible throughout the entire force increase applied by the tester. The handheld device recorded simultaneously the reaction force between tester and participant and the position of the limb during the MMT for objective evaluation.

### 2.7. Procedure

Prior to the measurement day, the participant received information on the study, the consent form and the access to the first questionnaire via email. On the measurement day, the tester introduced the participant to the procedure and the informed declaration of consent was signed before the second questionnaire was filled in. If mood and physical well-being were rated on the scale as >0, the preliminary MMTs of the elbow and hip flexors were executed on both sides (without handheld device). In case of regular stability, the participant was included, and the muscles of her preferred side were measured. If only one side showed regular stability, this side was used for the following measurements. Subsequently, the AF measurements followed. In total, 16 trials were performed, starting with two baseline measurements for each muscle (elbow and hip flexors). Twelve single-blinded, randomized measurements including the imagery tasks followed: each muscle (elbow and hip flexors) was tested three times (M1–M3) during pleasant and unpleasant imagery. The order of muscles and imagery being tested was randomized in Microsoft Excel (Microsoft 365) prior to the measurements. Double-blinding was not possible, since the participants had to actively imagine the food. To ensure single-blinding, the tester left the room while the participant was instructed to enter the imagery by the second assistant using the words noted earlier. The assistant read the words in a neutral and calm manner repeatedly for ~20 s and held the tablet with the chosen picture of the respective food straight above the participant’s eyes at the height of the hanging rail so that the participant kept a supine position without head rotation. Meanwhile, the other assistant prepared the handheld device and tablet for the new trial and recorded any relevant details such as noises. As soon as the participant successfully imagined the respective food (~20 s; indicated by head-nodding), the tester was informed to enter the laboratory. She had no eye contact, neither with the participant (avoided by the curtain) nor the assistants. The first assistant named the muscle to be tested and confirmed that the handheld device was ready for recording (“ok”) without making eye-contact. Any further verbal interaction was prohibited. The participant was previously instructed to focus on her imagination during the MMT. The whole procedure of one trial lasted ~40 s. Resting periods were ~60 s. After each trial, the result of the subjective rating of the MMT (stable/unstable) was given by the tester to the first assistant by a thumb up (stable) or thumb down (unstable) sign without eye contact.

After completing all AF measurements, both muscles were tested again for each imagery without blinding, without the curtain and without the handheld device. This checked the tester’s evaluation of the MMTs during pleasant and unpleasant imagery comparing blinded and unblinded trials. Interactions between all persons were allowed again. Subsequently, the participant was asked for feedback. Self-reported information on the imagery process or thoughts during the measurement were of interest and were recorded.

### 2.8. Data Processing and Statistical Analysis

The NI^TM^ DIAdem 2021 (National Instruments, Austin, TX, USA) was used for data processing. The csv-files of the recorded measurements were transferred from the tablet to NI^TM^ DIAdem 2021. The force and gyrometer signals were used for evaluation and were first checked visually, which led to the exclusion of trials in some cases.

#### 2.8.1. Exclusion of Trials

Sixteen trials were performed per participant (in total 192 trials). Only 161 trials were evaluated (elbow: 78; hip: 83) due to the following reasons: the elbow flexors of two participants and the hip flexors of one participant showed a dysfunction in the preliminary MMT and, therefore, those 24 trials were excluded from evaluation. So as not to change the measuring procedure, they were also measured and not omitted before. Furthermore, one trial of hip flexors had to be excluded because the tester contacted the knee with her chin during the MMT, which might have led to confusion of the tester and/or participant. Visual inspection led to exclusion of further six trials. In one trial a clear pushing by the participant against the tester was visible. This was not allowed since the participants should only perform holding isometric muscle actions. In further five trials the recording stopped before the end of the measurement. It is not clear if the tester pushed the stop button too early or if the measurement stopped because of technical issues.

#### 2.8.2. Data Processing and Relevant Parameters of Adaptive Force

The force and gyrometer signals of the 161 included trials were interpolated (linear spline interpolation) to gain equidistant time intervals (1000 Hz) and were filtered (Butterworth, cutoff frequency 20 Hz, filter degree 5) in NI^TM^ DIAdem 2021. The following parameters of interest were extracted for each trial of baseline, pleasant and unpleasant imagery (analogues to [11,41]):Maximal adaptive force (AF_max_)

The AF_max_ (N) refers to the maximal value of AF in the force curve that was reached during the entire trial, irrespective of whether the muscle lengthened or not. Thus, it can arise during isometric or eccentric muscle action. If muscle length stayed stable during the whole measurement, AF_max_ was reached under isometric conditions (AFiso_max_); if the muscle gave way, AF_max_ was obtained during eccentric muscle action (AFecc_max_). AF_max_ does not display the participant’s maximal force in general. Under isometric conditions, the AF_max_ depends also on the force applied by the tester, whereas in case of muscle lengthening, the AF_max_ is equal to the maximal eccentric force in the present circumstances and is less dependent on the tester’s force application.

2.Maximal isometric adaptive force (AFiso_max_)

This parameter is the most important one for the present investigation. AFiso_max_ (N) refers to the maximal AF, which was reached under isometric conditions. In case of muscular stability, AF_max_ = AFiso_max_. In case of instability, it marks the breaking point in which the participant’s muscle merges from isometric to eccentric action. To identify AFiso_max_, the gyrometer signal (deviation of angle over time) was analyzed. It oscillates around zero under isometric conditions but increases above zero as the muscle starts to lengthen. If the gyrometer signal increased above zero, the force value at the moment of last zero crossing of the gyrometer signal was referred to as AFiso_max_ (breaking point). If the gyrometer signal oscillated around zero throughout the entire force increase, AFiso_max_ = AF_max_.

The AFiso_max_ was furthermore related to AF_max_: $\frac{{\mathrm{AFiso}}_{\max}}{{\mathrm{AF}}_{\max}}$ (%). This should reflect the maximal holding capacity in relation to the maximal reached force value of the respective trial. Since AF_max_ does not necessarily reflect the maximal strength of the participant, a second ratio ($\frac{{\mathrm{AFiso}}_{\max}}{{\mathrm{maxAF}}_{\max}}$ (%)) was calculated additionally, whereby maxAF_max_ refers to the highest value of AF_max_ across all trials of the respective muscle and participant, irrespective if it was reached under isometric or eccentric conditions. Hence, maxAF_max_ was closest to the participant’s maximal strength.

3.Adaptive force at the onset of oscillations: AFosc

During the muscular interaction, an oscillating force equilibrium arises, especially in case of muscular stability. It is accompanied by emerging oscillations in the force signal, indicating a clearly distinguishable regular oscillatory behavior (swing up). It was assumed that this might be a prerequisite for muscular stability during the MMT [11,41]. Therefore, the force at the moment of onsetting oscillations (AFosc (N)) was investigated. The force signal was checked for oscillations (force maxima) appearing sequentially after the exponential phase of MMT (phase 2). If four force maxima with a time distance d_x_ < 0.15 s appeared consecutively, the force value of the first maximum was defined as AFosc, which marked the force at the onset of oscillations. d_x_ < 0.15 s was chosen since mechanical muscle oscillations occur around ~10 Hz [36,44,45,47,48]. If no such oscillations appeared during the entire trial, AFosc = AF_max_. For AFosc, the ratios to AF_max_ (%) and to maxAF_max_ (%) were calculated, too, to receive information on the force proportion at which oscillations arose regularly. It was previously found that AFosc appeared on a lower level than AFiso_max_, thus, on a lower level than AFiso_max_ in stable and on a higher level than AFiso_max_, in unstable MMTs. Therefore, the ratio $\frac{\mathrm{AFosc}}{{\mathrm{AFiso}}_{\max}}$ (%) was additionally calculated. This might give further insights into the neurophysiological relevance of those oscillations with regard to adaptation to external forces.

4.Slope of force rise

The force increase during MMT might affect the outcome. In particular, a steeper force rise could compromise the participant’s ability to stabilize the limb’s position. Therefore, similar slopes of force increase were a necessary prerequisite to compare the AF parameters between baseline, pleasant and unpleasant food imagery. Hence, the slope (N/s) of force increase was considered. For that, the arithmetic mean of the AFiso_max_ values of all as ‘unstable’ assessed trials of the respective muscle of one participant served as reference. The slope of each force curve (stable/unstable) was calculated by the difference quotient including the time and force values at 70% and 100% of this reference value (averaged AFiso_max_ value of all unstable trials). The decadic logarithm was applied to obtain the logarithmic slope (lg(N/s)) since the force rise was exponential. If the reference value occurred after the linear phase (transition to force plateau), the trial was excluded from slope analysis to avoid distortion. This was the case in 19 of 161 trials.

For the subsequent statistical evaluation, the arithmetic means (M), standard deviations (SD) and 95% confidence intervals (CIs) were calculated for each parameter separately per participant, muscle (elbow and hip flexors) and baseline or imagery (pleasant/unpleasant) and were used for statistical comparisons.

#### 2.8.3. Statistical Analyses

In total, ten complete datasets for elbow flexors and eleven for hip flexors were analyzed comparing baseline, pleasant and unpleasant imagery. Statistical analyses were performed using IBM SPSS Statistics (Windows, Version 28.0. Armonk, NY, USA: IBM Corp). The normal distribution of each parameter for each muscle in each condition (baseline, pleasant, unpleasant imagery) was checked by Shapiro–Wilk test. In case of normal distribution, repeated measures of ANOVA were performed (RM ANOVA). If sphericity was not given (Mauchly test), the Greenhouse–Geisser correction was chosen (F_Green_). For post hoc testing, Bonferroni correction was applied (adjusted *p* values are given by *p_adj_*). The effect size eta squared (*η*^2^) was calculated in IBM SPSS Statistics. For pairwise comparisons, the effect size Cohen’s d was calculated by *d_z_* $=\frac{\left|\mathrm{MD}\right|}{{\mathrm{SD}}_{\mathrm{MD}}}$, where MD stands for the mean difference and SD_MD_ for its standard deviation. The effect size was interpreted as small (0.2), moderate (0.5), large (0.80) or very large (1.3) [49,50]. Because RM ANOVA is considered to be robust against violation of normal distribution [51,52], the Friedman test to compare baseline, pleasant and unpleasant imagery was only used if more than one group was not normally distributed. This was the case for the parameters slope and $\frac{{\mathrm{AFiso}}_{\max}}{{\mathrm{AF}}_{\max}}$ for elbow and hip flexors, respectively. Kendall’s W was then calculated as effect size. Bonferroni post hoc testing was applied for pairwise comparisons (adjusted *p* values are given by *p_adj_*) and for effect size, Pearson’s *r* was calculated by *r* = $\left|\frac{z}{\sqrt{n}}\right|$. Significance level was set at α = 0.05.

## 3. Results

### 3.1. Rating of the MMT by the Tester

The single ratings of the tester for each trial and participant are given in Appendix A. The relative shares of the qualitative MMT ratings (stable vs. unstable) are visualized in Figure 2. For elbow flexors, all of the 19 baseline trials were rated as stable (100%); for pleasant imagery, 26 of 29 trials were assessed as stable (90%) and three as unstable (10%); for unpleasant imagery, 3 of 30 trials were rated as stable (10%) and 27 as unstable (90%). For hip flexors, 21 of 22 of baseline trials were rated as stable (95%), one as unclear (5%); 26 of 30 MMTs during pleasant imagery were rated as stable (87%) and 4 as unstable (13%); 2 of 32 MMTs during unpleasant imagery were rated as stable (6.25%), 27 as unstable (84.38%) and three as unclear (9.38%). The subsequent statistical evaluation was only based on the grouping of conditions (baseline, pleasant, unpleasant), independent of the tester’s rating.

The comparison of the MMT ratings of blinded vs. unblinded trials (without handheld device after AF assessment) showed in most cases that the imagery led to the expected MMT outcome (stable for pleasant, unstable for unpleasant imagery). For elbow flexors, all 20 unblinded MMTs were rated accordingly as stable during pleasant imagery (100%). During unpleasant imagery, the elbow flexors of one participant remained stable for one of the two unpleasant food visualizations (5%), which was in contrast to the hypothesis. All other 19 unblinded MMTs during unpleasant imagery were rated as unstable (95%), according to the hypothesis. For hip flexors, 20 of the 22 MMTs showed stability during pleasant imagery (91%), the remaining two were rated as unclear (9%) and pertained to the same participant. During unpleasant imagery, 21 of 22 unblinded MMTs of hip flexors were assessed as unstable (95.5%), according to the hypothesis; one was rated as unclear (4.5%).

### 3.2. Quality of Imagery

Four participants reported to have difficulties imagining the food experiences: for unpleasant ones in general, with repetition of the imagery, the alternation between pleasant and unpleasant imagery or because of discrepancies between the assistant’s instructions and own imagination. In all of those four participants, the result of at least one trial was not according to the hypothesis that AF would be stable during pleasant imagery and unstable during unpleasant imagery (refers to 26% of all trials of those four participants). One of the participants with the highest number of discrepancies between expected and occurred result of MMTs (50% according to hypothesis) showed insecurities during the test by, e.g., excusing herself if the muscle was not stable. It seems that she was irritated by the outcome of MMTs in the course of the measurements, especially in case of instability. Two further participants had difficulties with the imagery. They also showed deviations from the hypothesis in 3 of 12 MMTs for hip flexors (25%). Only two of the remaining 6 participants who reported no difficulties in imagining the food experience had one trial each involving the elbow flexors which did not show results according to the hypothesis; thus, the MMT results of those six participants without imagery difficulties were according to the hypothesis in 97% of all trials. This might indicate that the trials which were not assessed according to the hypothesis could result from the imagery quality rather than from the tester’s MMT performance.

### 3.3. Exemplary Force and Gyrometer Signals

Figure 3 exemplifies the force and gyrometer signals of elbow and hip flexors of one participant for baseline, pleasant and unpleasant imagery. The MMTs were rated as stable for baseline and pleasant imagery and as unstable for unpleasant ones. The force curves show nearly identical slopes, especially at the beginning, which is considered as the crucial phase for adaptation [11,41,43]. This reflects the high reproducibility of the tester’s force application.

The gyrometer signals during unpleasant imagery (Figure 3, red) showed an increase above zero. The related force values at those breaking points marked the maximal holding capacity for unpleasant imagery (AFiso_max_; elbow: 100 N, hip: 103 N), which was considerably lower than the AF_max_ of those trials, which were reached during muscle lengthening (AF_max_ = AFecc_max_; elbow: 138 N, hip: 166 N). Thus, the muscle started to lengthen at 73% and 62%, respectively, of the maximal force capacity. This reflects an inappropriate adaptation. In contrast, the gyrometer signals during baseline (grey) and pleasant imagery (blue) oscillated around zero throughout the entire force increase, reflecting the quasi-isometric muscle state until the maximum was reached (AF_max_ = AFiso_max_; elbow: 153 N and 143 N; hip: 185 N and 190 N). Minor liftoffs were related to the slight muscle suspension during MMT, which were accepted due to the freely moveable limb [41]. Consequently, for baseline and during pleasant imagery muscle lengthening did not occur. However, the same participant was not able to reach her maximal holding capacity under the influence of unpleasant imagery: the muscle gave way during force increase, resulting in a considerably lower AFiso_max_ compared to baseline and pleasant imagery (−32% (elbow) and −45% (hip)). However, the force increased further during muscle lengthening. It is notable for this example that even AF_max_ was slightly lower for unpleasant vs. baseline and pleasant imagery (elbow: ~93 ± 5%; hip: ~89 ± 2%).

The onset of oscillations did not appear as clear as hypothesized on the basis of the previous studies. The AFosc during unpleasant imagery was 138 N and 153 N for elbow and hip flexors, respectively. For baseline and during pleasant imagery, the AFosc amounted to 133 N and 130 N for elbow and 149 N and 190 N for hip flexors, respectively. Nevertheless, for unpleasant imagery the onset of oscillations appeared clearly after the breaking point, thus, during muscle lengthening; whereas for baseline and pleasant imagery, it occurred under isometric conditions (before AFiso_max_ was reached). Exempted from this was the MMT of hip flexors during pleasant imagery, whereby oscillations rose simultaneously to AF_max_. This will be discussed later. The results of this example are supported by the following statistical group comparisons. The single values of all parameters of each trial, muscle and participant are given in Appendix A.

### 3.4. Slope of Force Rise

The prerequisite of similar slopes of force rises for elbow and hip flexors is given indicated by a non-significant difference between baseline, pleasant and unpleasant imagery (elbow: χ^2^(2) = 0.722, *p* = 0.697, *n* = 10; hip: χ^2^(2) = 0.250, *p* = 0.882, *n* = 8, Figure 4, Table 1, Appendix A). Hence, the requirement of reproducible force profiles for comparing the AF parameters between the three conditions was fulfilled.

### 3.5. Maximal Adaptive Force of Elbow and Hip Flexors

For elbow flexors, AF_max_ showed no significant differences between the three conditions (F(2,18) = 0.683, *p* = 0.518), reflecting that AF_max_, irrespective if reached during isometric or eccentric muscle action, was similar between baseline, pleasant and unpleasant imagery (Figure 5a, Table 1 and Appendix A). AF_max_ during pleasant imagery was ~5.5 ± 16.6% higher and during unpleasant imagery ~12.9 ± 30.4% higher than baseline, respectively. AF_max_ during unpleasant imagery was ~6.6 ± 22.1% higher than AF_max_ during pleasant imagery.

For hip flexors, RM ANOVA was significant (F(2,20) = 4.777, *p* = 0.020, *η*^2^ = 0.323) (Figure 5d, Table 1 and Appendix A). Pairwise comparisons revealed a just significantly lower AF_max_ for baseline vs. unpleasant imagery (t(10) = −2.928, *p_adj_* = 0.045, *d_z_* = 0.883). Differences between baseline vs. pleasant and unpleasant vs. pleasant imagery were non-significant (*p_adj_* = 0.498 and *p_adj_* = 0.379, respectively). AF_max_ during pleasant imagery was ~9.46 ± 18.05% higher and during unpleasant imagery ~14.84 ± 14.03% higher than AF_max_ of baseline. During unpleasant imagery, AF_max_ was ~4.13 ± 13.96% higher compared to pleasant imagery.

Since AF_max_ for unpleasant imagery was always reached during eccentric muscle action, the decisive parameter for comparison was AFiso_max_, which was reached during isometric action in each condition (baseline, pleasant, unpleasant).

### 3.6. Maximal Isometric Adaptive Force of Elbow and Hip Flexors

For elbow and hip flexors AFiso_max_ was significantly lower during unpleasant vs. pleasant imagery and vs. baseline (Table 1, Figure 5b,e, Appendix A). No significant difference was found between baseline and pleasant imagery. The ratio of AFiso_max_ to AF_max_ showed that during unpleasant imagery, the muscle started to lengthen at a clearly lower AF level than during pleasant imagery or compared to baseline, respectively (Table 1, Figure 5c,f). Consequently, $\frac{{\mathrm{AFiso}}_{\max}}{{\mathrm{AF}}_{\max}}$ differed significantly in Friedman test between the three conditions for both muscles. Bonferroni post hoc testing revealed a significant difference between baseline vs. unpleasant imagery (elbow: *z* = 1.650, *p_adj_* = 0.001, *r* = 0.52, *n* = 10; hip: *z* = 1.636, *p_adj_* < 0.001, *r* = 0.49, *n* = 11) and for pleasant vs. unpleasant imagery (elbow: *z* = 1.350, *p_adj_* = 0.008, *r* = 0.43, *n* = 10; hip: *z* = 1.364, *p_adj_* = 0.004, *r* = 0.41, *n* = 11). Baseline vs. pleasant imagery differences were non-significant (elbow: *z* = 0.300, *p_adj_* = 1.000; hip: *z* = 0.273, *p_adj_* = 1.000).

The overall maximal value of AF (maxAF_max_) was on average 202.43 ± 31.31 N for elbow (n = 10) and 216.82 ± 25.67 N for hip flexors (n = 11). Taking the individual maxAF_max_ values (Appendix A) as references, the AFiso_max_ of elbow flexors amounted ~78.31 ± 16.50% of maxAF_max_ for baseline and ~77.79 ± 10.95% for pleasant imagery, which was significantly higher than for unpleasant imagery (~44.79 ± 16.54%; F(2,18) = 28.105, *p* < 0.00001, *η*^2^ = 0.757); similar for hip flexors: $\frac{{\mathrm{AFiso}}_{\max}}{{\mathrm{maxAF}}_{\max}}$ = 73.34 ± 11.25%, 73.00 ± 14.35% and 52.76 ± 19.78 % for baseline, pleasant and unpleasant imagery, respectively (F(2,20) = 11.678, *p* < 0.001, *η*^2^ = 0.539). For both muscles, pairwise comparisons showed significantly lower $\frac{{\mathrm{AFiso}}_{\max}}{{\mathrm{maxAF}}_{\max}}$ values for unpleasant vs. pleasant imagery (elbow: t(9) = −5.684, *p_adj_* = 0.001, *d_z_* = 1.798; hip: t(10) = −3.471, *p_adj_* = 0.018, *d_z_* = 1.047) as well as for unpleasant imagery vs. baseline (elbow: t(9) = −5.559, *p_adj_* = 0.001, *d_z_* = 1.758; hip: t(10) = −3.888, *p_adj_* = 0.005, *d_z_* = 1.172), whereby the difference was non-significant for pleasant imagery vs. baseline (both muscles: *p_adj_* = 1.000).

The results clearly show that the participants were not able to appropriately adapt their muscles in an isometrically holding manner to the external force increase while imagining unpleasant food experiences. The maximal holding capacity was reduced so that the muscles gave way at a significantly lower force compared to baseline and pleasant imagery. Comparing the results of AFiso_max_ to the tester’s MMT ratings, a high agreement is visible. The MMTs rated as stable showed a value of $\frac{{\mathrm{AFiso}}_{\max}}{{\mathrm{AF}}_{\max}}\;$= 100.00 ± 0.00% for elbow and 99.74 ± 0.87% for hip flexors, whereas for unstable rated MMTs it was 51.36 ± 18.25% and 58.34 ± 21.08%, respectively. This indicates the evaluation of the data of the handheld device supports the tester’s ratings.

### 3.7. Adaptive Force at the Onset of Oscillations

For elbow flexors, AFosc was significantly lower for baseline and for pleasant vs. unpleasant imagery (Figure 6, Table 1 and Appendix A), indicating an oscillatory swing up occurred at a considerably lower force level (baseline vs. unpleasant: t(9) = −5.747, *p_adj_* = 0.001, *d_z_* = 1.817, pleasant vs. unpleasant: t(9) = −5.457, *p_adj_* = 0.001, *d_z_* = 1.726). $\frac{\mathrm{AFosc}}{{\mathrm{AFiso}}_{\max}}$ demonstrated that the oscillations for baseline and pleasant imagery appeared during the force increase before AFiso_max_ was reached (averaged over both groups: 78.64 ± 13.62%), but during unpleasant imagery they appeared—if at all—after the breaking point for each participant (217.76 ± 108.72%).

For hip flexors, the results showed a slightly different behavior (Table 1 and Appendix A, Figure 6). RM ANOVA was significant in comparing baseline, pleasant and unpleasant imagery (F_Green_(1.19,11.85) = 5.804, *p* = 0.029, *η*^2^ = 0.367). Although AFosc was highest during unpleasant imagery, pairwise comparison of pleasant vs. unpleasant imagery did not differ significantly (Bonferroni correction: *p_adj_* = 0.591). Nevertheless, AFosc was significantly lower for baseline vs. unpleasant imagery (t(10) = −3.665, *p**_adj_* = 0.013, *d_z_* = 1.105). The values of the ratio AFosc to AFiso_max_ (Table 1) reflect the high variation of the oscillatory onset related to the breaking point for unpleasant imagery, which was also visible in the 95% CIs (Figure 6f). RM ANOVA comparing $\frac{\mathrm{AFosc}}{{\mathrm{AFiso}}_{\max}}$ values for baseline, pleasant and unpleasant imagery just missed significance after the required Greenhouse–Geisser correction (F_Green_(1.019,10.190) = 4.747, *p* = 0.053).

### 3.8. Case Example: Adaptive Force under Current Stress vs. Positive Imagery

One female participant (28 years, 174 cm, 69 kg) initially showed an impaired neuromuscular function in preliminary MMTs. A brief talk revealed she experienced current stress (time pressure, several exams ahead). The initial instability of her muscles could be remedied immediately by positive imagery (yoga or victory in kickboxing competition; both good feelings of relaxation and strength). AF measurements of elbow and hip flexors were performed using the negative imagery (exam situation) vs. the positive ones (yoga/kickboxing) randomized and single-blinded (each 3× per muscle, same setting as above without baseline). For negative imagery, the elbow flexors showed an unstable behavior in 2 of 3 trials; in one trial the MMT was rated as stable despite negative imagery. For positive imagery, the elbow flexors were rated as stable for all trials. For hip flexors, all three trials during negative imagery were unstable, for positive imagery, the first two trials were assessed as stable and the last one as unstable. Figure 7 displays exemplary measuring curves of force and gyrometer signals for elbow and hip flexors during positive (blue) and negative imagery (red). The tester’s MMT ratings were again confirmed by the recorded data of the handheld device.

The$\;\frac{{\mathrm{AFiso}}_{\max}}{{\mathrm{AF}}_{\max}}$ for elbow flexors was 62 ± 37% for negative and 100 ± 0% for positive imagery. For hip flexors it amounted to 40 ± 9% (negative) and 84 ± 27% (positive). The $\frac{\mathrm{AFosc}}{{\;\mathrm{AFiso}}_{\max}}$ for elbow flexors was 189 ± 177% and 49 ± 5% for negative and positive imagery, respectively, and for hip flexors 201 ± 25% and 93 ± 61%, respectively. Summarizing, the holding capacity (AFiso_max_) was considerably reduced during negative imagery, which was also the entry state of the participant. By imagining positive experiences, the holding capacity was immediately improved, indicating a regular neuromuscular function. The oscillatory onset in stable MMTs occurred before and for unstable MMTs after AFiso_max_ was reached.

## 4. Discussion

The aim of the present study was to investigate the influence of pleasant and unpleasant emotional imagery on different AF parameters of elbow and hip flexors in an improved design including randomization, single-blinding and baseline measurements. Since the force rises (slope) did not differ significantly between baseline, pleasant and unpleasant imagery, the subsequent discussion is based on reproducible force rises and, thus, similar performances of the tester’s force application.

It was hypothesized the MMTs would be rated as stable for baseline and pleasant imaginations and as unstable for unpleasant ones. The ratings of blinded MMTs did not differ considerably compared to the non-blinded ones assessed without handheld device after the objectified AF measurements. The missing blinding was the main limitation of the previous study. However, 88% of all 121 blinded MMTs (baseline measurements were excluded) and 95% of all 84 unblinded MMTs were rated according to the hypotheses in this study. Thus, even though the tester did not know which imagery was conducted, the subjective ratings of MMTs supported the hypothesis in the vast majority of trials. On that basis it can be stated that blinding did not lead to a considerably different MMT rating by the tester compared to non-blinding. Some limitations regarding imagery quality (mentioned above) must be considered which possibly led to the deviating results.

The main hypothesis of this study was supported by the results: the muscles started to give way at a clearly and significantly lower force (AFiso_max_) during unpleasant imagery compared to pleasant ones and baseline measurements. This result indicates that unpleasant imagery resulted in a worse adaptation capability in healthy participants, irrespective of the muscle. Under the assumption that unpleasant food imagery is related to the negative emotion disgust, this suggests that such negative emotions seem to affect the muscle function in the sense of isometric AF also in healthy participants. The value of this finding will be discussed later.

The second hypothesis was mainly confirmed by the results. The AF_max_ did not differ between baseline, pleasant and unpleasant imagery for elbow flexors. For hip flexors, however, the AF_max_ was just significantly higher for unpleasant imagery vs. baseline. Since the limb’s position was stable for all baseline measurements (AF_max_ = AFiso_max_), the lower AF_max_ was due to the lower maximally applied force by the tester. Therefore, the adaptive capability of the participant was not challenged by higher forces. However, one outlier existed which might have led to the significant result: the baseline AF_max_ of one participant was clearly lower compared to the respective AF_max_ during unpleasant imagery (116.66 N vs. 193.98 N). By excluding this participant, baseline vs. unpleasant imagery did not differ significantly (*p_adj_* = 0.096). As mentioned in the methods, the tester stopped the force increase if an oscillatory equilibrium between tester and participant was reached on a considerably high force level. Looking at the onset of oscillations for the baseline trials of this participant, oscillations had already appeared at 58.02 N, which was the lowest of all participants. Those oscillations might have caused a feeling of stability which, in turn, led to termination of force increase by the tester. The essential result was, however, that the AF under isometric conditions was significantly lower for unpleasant imagery, despite of AF_max_.

Regarding the third hypothesis (onset of oscillations), the results of AFosc of elbow flexors confirmed the first study, since oscillations appeared at a significantly higher AF during unpleasant vs. pleasant imagery and baseline; furthermore, during unpleasant imagery they appeared—if at all—after the breaking point for each participant, thus during muscle lengthening. For hip flexors, a deviating behavior was found for some participants. A trend according to the hypothesis was visible, e.g., AFosc for baseline was significantly lower than for unpleasant imagery. During pleasant vs. unpleasant imagery, AFosc differed insignificantly, however, the results were close to significance (*p_adj_* = 0.079). Nevertheless, three participants showed a low AFosc also during unpleasant imagery. It was visible, thereby, that the oscillations were not as clear and as constant as during pleasant imagery and at baseline. This might highlight some methodological limitations for AFosc evaluation.

### 4.1. Limitations

Although the design of the present study was improved compared to the previous one and included single-blinding, randomization and baseline measurements, some limitations especially regarding the evaluation should be considered. The evaluation of the oscillatory onset (AFosc) was adopted from the previous study and was based on the criterion that four consecutive maxima in force increases with a time interval d_x_ < 0.15 s must arise. This was chosen since mechanical muscular oscillations are known to show low frequencies around 10 Hz. The amplitude and concrete frequency of those oscillations were not considered. As mentioned above, in some cases a clear swing up was missing, although four consecutive maxima occurred. Therefore, the algorithm of oscillatory onset ought to be revised. From a visual inspection, the frequency and the amplitude seem to differ between stable and unstable MMTs. Therefore, probably a power-frequency analysis could be applied in further studies.

Another limitation might be that the quality of imagery was not assessed quantitatively. The self-reporting after the measurements gave an impression of how well the participant interpreted their ability to imagine the food experience. Some participants had difficulties imagining the food, as mentioned above. This might have led to deviations regarding the stability of AF. However, the aim was not to quantify the quality of imagination but their effect on AF in healthy individuals in general. Therefore, they were included.

Possible limitations concerning MMT performance were previously described [11,41], especially regarding the maximal value and slope of force application. The hip flexors showed a slightly but still significantly lower AF_max_ for baseline measurements vs. unpleasant imagery trials in this study (discussed above), which was not expected. This is presumably a result of the lower maximal force applied by the tester, probably because of perceiving the mutual oscillations already at a low force level. Since the crucial parameter was the AF under isometric conditions (AFiso_max_), which was still clearly and significantly lower during unpleasant imagery, the lower AF_max_ of hip flexors for baseline can be neglected here.

### 4.2. Neurophysiological Considerations and Practical Implications

A detailed proposed explanation of neurophysiological processing during the execution of AF as well as the effect of emotions were given previously [11,41,43]. It is assumed that the pleasant and unpleasant imagery trigger positive (pleasure) and negative emotions (disgust), respectively. This is in accordance with results of other authors [25,53]. The links between food and emotions were also discussed. Heldke wrote that “tastes (…) are like feelings” [54]. It is plausible that imagining a food experience triggers emotions. These are highly individual and are accompanied by the related memorable affective experiences.

Since several central structures are involved in processing motor control as well as emotions [5,6,7,8,9,10], the influence of emotions on motor control is conceivable. However, as mentioned in the introduction, few studies have investigated the influence of positive and negative emotions on muscular activity in healthy participants [12,25,32]. It was suggested that AF, especially the holding capacity, characterizes a particular functioning of neuromuscular control which seems to be highly sensitive to interfering inputs. Based on the findings, negative imagery apparently results in a substantial muscular instability even in healthy participants. It can only be assumed how stressful situations and traumas influence the AF. If stress is persisting (at work, in relationships, or conditions after traumatic experiences), we expect a permanently impaired holding function. This is based on our own experience of long-term clinical practice and is reinforced by the present findings. The effect of unpleasant food imagery was only temporary and could be reversed immediately by imagining pleasant experiences. We hypothesize a positive effect on muscular stability by imagining positive situations in persons suffering from actual or chronic stress. The presented case example supports those hypotheses, since the participant showed a muscular instability in her entry state, reporting currently perceived mental stress. Positive imagery improved the participant’s AFiso_max_ immediately. Based on the findings and the practical experience, we propose AF assessment could be suitable to evaluate the positive and negative effects of emotions. Moreover, it seems to be appropriate to test which imagery can improve the holding capacity since this particular muscular function can instantaneously change. It can only be assumed that other bodily systems—autonomous nervous, endocrine and immune systems—behave similarly and would switch from a dysfunction to normal regulation. The underlying potential of such an approach can be imagined thereby and might have applications in rehabilitative and clinical settings. Since not only emotional states are affecting motor processing but also various afferences such as, e.g., nociception [55,56], it is suggested that the AF could be also influenced by other disturbing factors.

In daily activities and sports, the adaptive capacity of the neuromuscular system is necessary for joint stabilization. If it is reduced, joints could suffer from inappropriate joint alignment under strain, which might result in pain and probably leads to degeneration or increased risk of injury. This could explain the still poorly understood “overuse” injuries and might clarify the causal chain regarding the joint appearance of musculoskeletal pain, muscle weakness and mental stress. From our point of view, mental issues lead to an impaired neuromuscular function which, in turn, can result in complaints of the musculoskeletal system.

### 4.3. Characterization of ‘Stable’ vs. ‘Unstable’ Adaptation

In the previous studies investigating the AF behavior regarding pleasant and unpleasant imagery/odors [11,41], concrete values characterizing stable and unstable adaptation were proposed. Including the findings of this study, the suggested values can be extended. In the following, the data of all three studies are included. Stable adaptation seems to be characterized by a high AFiso_max_ ≈ AF_max_ (≥ 99% of AF_max_), indicating the muscle length stays quasi-isometric during the entire force increase. Unstable adaptations, in turn, show a significantly lower AFiso_max_ ≈ 56% of AF_max_, reflecting that the muscle starts to lengthen at a significantly lower force during adaptation. Furthermore, during stable adaptation oscillations occur at a force level of ~74% of AF_max_; in turn, for unstable adaptation they appear at ~88% of AF_max_. In the previous studies and for elbow flexors in the present study, the onset of oscillations for unstable adaptation was at a higher force level (~95%). It is not clear why the oscillations under unstable conditions for hip flexors arose at a low force level (~75%) in the present study. As mentioned, the evaluation or possibly the participant’s regulatory state might play a role. However, it seems to be more important if oscillations arise before or after the breaking point ($\frac{\mathrm{AFosc}}{{\mathrm{AFiso}}_{\max}}$). Considering the data of all three studies, 183 stable and 124 unstable MMTs were recorded during AF assessment. In 177 stable trials (96.9%) oscillations occurred before AFiso_max_, thus, under isometric conditions; in 108 unstable trials (87.4%) the oscillations arose after the breaking point, thus, during muscle lengthening. The occurrence of those regular oscillations might be a prerequisite for stability during muscular adaptation. Further investigations are necessary to verify this hypothesis.

## 5. Conclusions

The present study investigated the motor adaptation in the sense of AF in reaction to emotional imagery in a single-blinded, randomized setting. The results support the previous findings: the maximal holding capacity (AFiso_max_) was reduced highly significantly and instantaneously by imagining unpleasant food experiences. The conclusion is that negative emotions, such as disgust, can lead to muscular instability. Assuming this might result in joint destabilization under strain, it could pave the way for explaining the causal chain regarding the link between musculoskeletal pain and mental health states. It is proposed that an impaired holding function due to mental stress could lead to musculoskeletal pain.

Investigating the adaptive holding function might be a promising innovative approach to obtain insights into psychomotor states and to support diagnostics of mental health conditions. AF might be used to test instantaneously the effect of emotions on motor function. This immediate change of muscular stability as reaction to positive or negative stimuli might further help to determine purposeful therapies. This might open up innovative and highly beneficial possibilities regarding psychological diagnostic and treatment approaches. Further research is needed to examine this hypothesis.

Moreover, the collected data until now suppose that a proper ‘regular’ neuromuscular adaptation might be characterized by oscillations. This provides novel insights into neuromuscular control. Further research is needed to investigate if they could be a prerequisite for reacting and adapting adequately to external forces.

## Figures and Tables

**Figure 1 brainsci-12-01318-f001:**
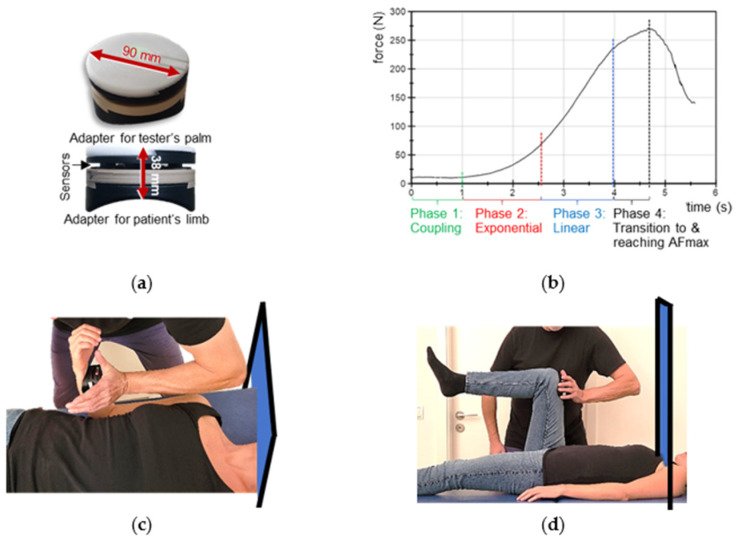
Handheld device, optimal force profile and setting. (**a**) Handheld device. (**b**) Suggested optimal force profile applied externally by the tester during the MMT, which consists of the four illustrated phases (according to Bittmann et al. [43]). (**c**) Starting position of the MMTs of elbow flexors and (**d**) of hip flexors including the curtain which prevented visual contact between tester and participants during the trials.

**Figure 2 brainsci-12-01318-f002:**
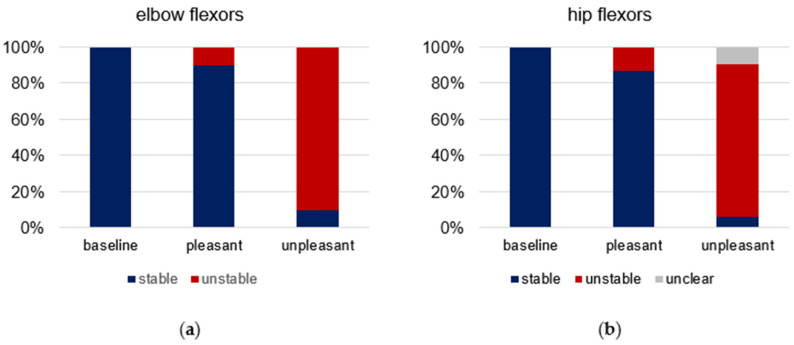
Rating of manual muscle tests (MMT) by the tester. Displayed are the relative shares of the qualitative rating of all MMTs (stable = blue, unstable = red, unclear = grey) regarding baseline (elbow: *n* = 19, hip: *n* = 21), pleasant (elbow: *n* = 29, hip: *n* = 30) and unpleasant (elbow: *n* = 30, hip: *n* = 32) imagery for (**a**) elbow and (**b**) hip flexors.

**Figure 3 brainsci-12-01318-f003:**
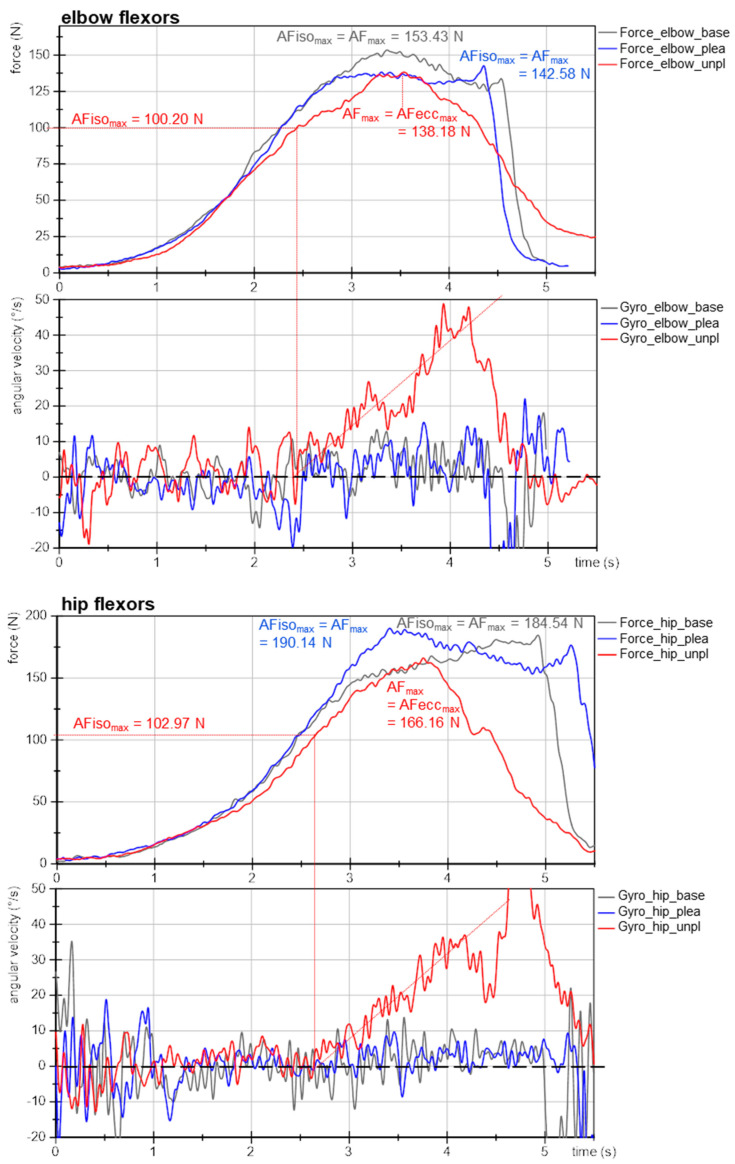
Exemplary signals of Adaptive Force during MMT. Displayed are force (N) and gyrometer signals (°/s) during MMT of elbow and hip flexors of the same participant (age: 22 years, height: 171 cm, body mass: 63 kg) for baseline (grey, base), pleasant (blue; plea) and unpleasant (red; unpl) imagery. Parameters AF_max_, AFecc_max_ and AFiso_max_ are marked.

**Figure 4 brainsci-12-01318-f004:**
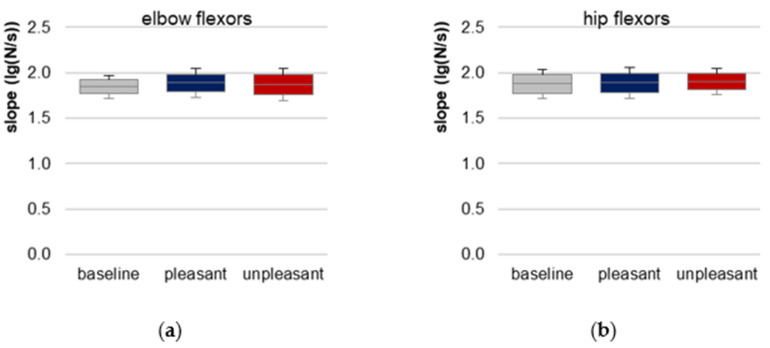
Slope. Arithmetic means, standard deviations (error bars) and 95% confidence intervals (CIs) of the logarithmic slope (lg(N/s)) comparing baseline (grey), pleasant (blue) and unpleasant (red) imagery of (**a**) elbow and (**b**) hip flexors are displayed. Statistical comparisons were non-significant (*p* > 0.05).

**Figure 5 brainsci-12-01318-f005:**
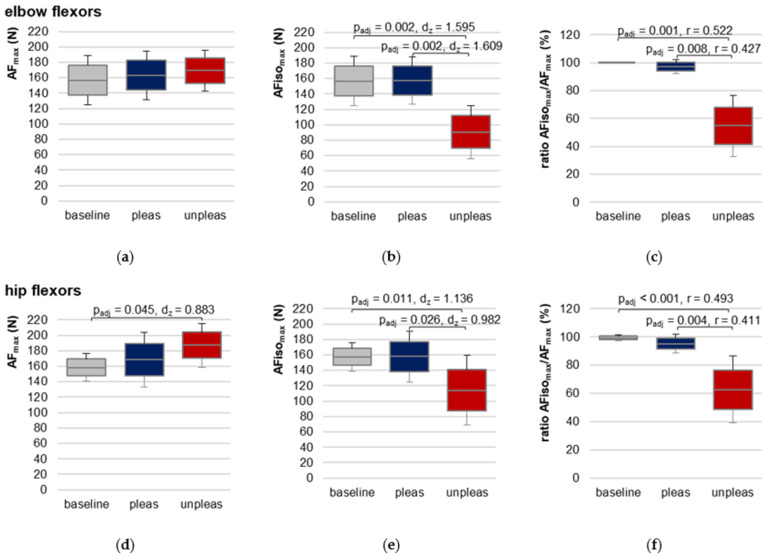
Maximal Adaptive Force and maximal isometric Adaptive Force. Arithmetic means, standard deviations (error bars) and 95% CIs of the maximal Adaptive Force (AF_max_; (**a**,**d**)), the maximal isometric Adaptive Force (AFiso_max_; (**b**,**e**)) and their ratio (**c**,**f**) compared between baseline (grey), pleasant (pleas, blue) and unpleasant (unpleas, red) imagery for elbow (**a**–**c**) and hip flexors (**d**–**f**) are displayed. Adjusted *p* values (Bonferroni correction) and effect sizes (Cohen’s *d_z_* or Pearson’s *r*) are given in case of significance.

**Figure 6 brainsci-12-01318-f006:**
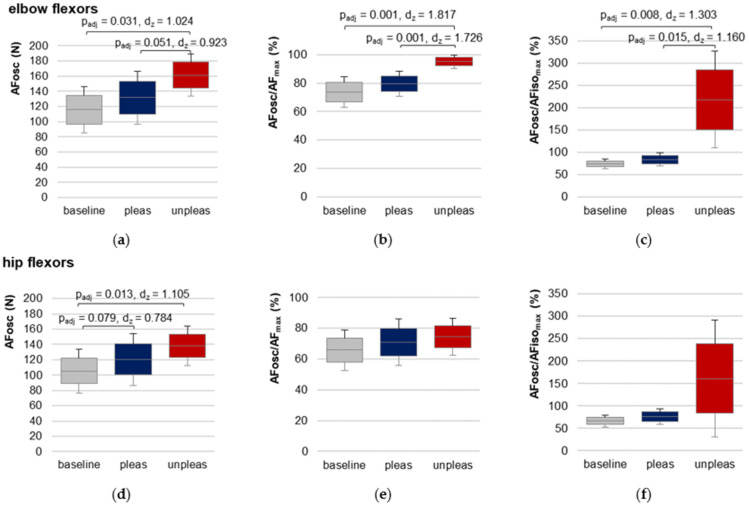
Adaptive Force at the onset of oscillations. Arithmetic means, standard deviations (error bars) and 95% CIs of the AF at the onset of oscillations (AFosc (N); (**a**,**d**)) and the ratios AFosc to AF_max_ (%) (**b**,**e**) and to AFiso_max_ (%) (**c**,**f**) compared between baseline (grey), pleasant (pleas, blue) and unpleasant (unpleas, red) imagery for elbow flexors (**a**–**c**) and hip flexors (**d**–**f**) are displayed. Adjusted *p* values (Bonferroni correction) and effect sizes Cohen’s *d_z_* are given in case of significance.

**Figure 7 brainsci-12-01318-f007:**
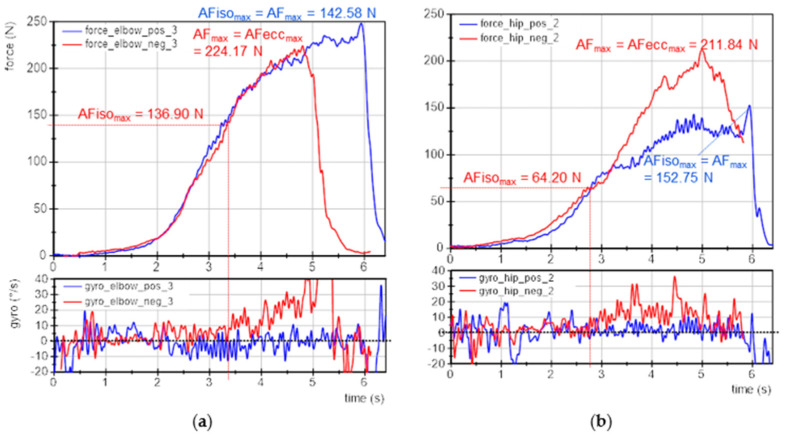
Case example: exemplary signals of objectified MMT. Displayed are the force (N) and gyrometer signals (°/s) during MMTs of (**a**) elbow and (**b**) hip flexors of the same participant (age: 27 years, height: 174 cm, body mass: 69 kg) during positive (blue; pos) and negative (red; neg) imagery. Marked are the parameters AF_max_, AFecc_max_ and AFiso_max_.

**Table 1 brainsci-12-01318-t001:** Descriptive and statistical results of AF parameters. Arithmetic means (M), standard deviations (SD), lower and upper border of 95% CIs as well as *p* values and effect sizes *η*^2^ or Kendall’s W of the AF parameters for baseline, pleasant and unpleasant imagery of elbow and hip flexors are given.

Parameter	Imagery	M ± SD	Borders of 95%-CI	Significance *p*	*η*^2^ or Kendall’s *W* ^b^
Elbow flexors					
AF_max_ (N)	baseline	156.87 ± 31.83	137.15; 176.60	0.518	-
	pleasant	163.15 ± 31.39	143.70; 182.61		
	unpleasant	169.26 ± 26.56	152.80; 185.73		
AFiso_max_ (N)	baseline	156.87 ± 31.83	137.15; 176.60	**<0.0001**	0.720
	pleasant	157.20 ± 30.38	138.38; 176.03		
	unpleasant	90.45 ± 34.20	69.25; 111.64		
Ratio AFiso_max_ to AF_max_ (%)	baseline	100 ± 0	-	**<0.0001 ^b^**	0.936 ^b^
	pleasant	97.14 ± 5.00	94.04; 100.23		
	unpleasant	54.82 ± 21.74	41.34; 68.29		
AFosc (N)	baseline	115.93 ± 30.33	97.14; 134.73	**0.003**	0.478
	pleasant	131.64 ± 34.69	110.14; 153.14		
	unpleasant	161.54 ± 27.54	144.47; 178.62		
Ratio AFosc to AF_max_ (%)	baseline	73.67 ± 10.87	66.94; 80.41	**<0.0001**	0.720
	pleasant	79.59 ± 8.88	74.08; 85.09		
	unpleasant	95.16 ± 4.63	92.29; 98.03		
Ratio AFosc to AFiso_max_ (%)	baseline	73.67 ± 10.87	66.94; 80.41	**0.003 ^a^**	0.626
	pleasant	83.62 ± 14.79	74.45; 92.78		
	unpleasant	217.75 ± 108.72	150.37; 285.14		
Slope lg(N/s)	baseline	1.85 ± 0.13	1.77; 1.93	0.836 ^b^	-
	pleasant	1.89 ± 0.16	1.79; 1.98		
	unpleasant	1.87 ± 0.17	1.76; 1.98		
Hip flexors					
AF_max_ (N)	baseline	158.51 ± 18.02	147.86; 169.16	**0.020**	0.323
	pleasant	168.62 ± 35.01	147.93; 189.31		
	unpleasant	187.02 ± 28.58	170.13; 203.90		
AFiso_max_ (N)	baseline	157.38 ± 18.24	146.60; 168.16	**<0.001**	0.512
	pleasant	157.89 ± 33.13	138.31; 177.47		
	unpleasant	114.20 ± 45.28	87.44; 140.96		
Ratio AFiso_max_ to AF_max_ (%)	baseline	99.35 ± 2.17	98.06; 100.63	**<0.0001 ^b^**	0.890 ^b^
	pleasant	95.22 ± 6.66	91.29; 99.15		
	unpleasant	62.80 ± 23.53	48.90; 76.71		
AFosc (N)	baseline	105.51 ± 28.49	88.67; 122.35	**0.029 ^a^**	0.367
	pleasant	120.41 ± 34.05	100.29; 140.53		
	unpleasant	137.99 ± 25.86	122.71; 153.27		
Ratio AFosc to AF_max_ (%)	baseline	65.83 ± 13.06	58.11; 73.55	0.168	-
	pleasant	70.76 ± 15.18	61.79; 79.73		
	unpleasant	74.40 ± 12.18	67.20; 81.60		
Ratio AFosc to AFiso_max_ (%)	baseline	66.40 ± 13.30	58.54; 74.26	0.053 ^a^	-
	pleasant	76.27 ± 17.50	65.93; 86.62		
	unpleasant	161.17 ± 130.54	84.02; 238.31		
Slope lg(N/s)	baseline	1.88 ± 0.16	1.77; 1.98	0.687 ^b^	-
	pleasant	1.89 ± 0.17	1.79; 2.00		
	unpleasant	1.90 ± 0.15	1.82; 1.99		

^a^ Greenhouse–Geisser correction; ^b^ Friedman test incl. Kendall’s W. Significant results are displayed in bold.

## Data Availability

The data presented in this study are available in article and Appendix A.

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
