# Peer review of "Emotional Imagery Influences the Adaptive Force in Young Women: Unpleasant Imagery Reduces Instantaneously the Muscular Holding Capacity"

_brainsci, 2022, doi:10.3390/brainsci12101318_

Round 1
Reviewer 1 Report
I have read with a lot of interest manuscript entitled: 'Emotionally affective imaginations influence the Adaptive Force in young women: unpleasant imaginations reduce instantaneously the muscular holding capacity'. Authors present a study (I think it is enough to say either Emotional imagery or Affective imagery, not both) on the adaptive force in a well controlled, single-blinded study, with 12 females. Participants were asked to generate AF of elbow/hip flexors quantified by handheld device for the manual muscle test. During the task participants perfomed the emotional imagery task of their preferred food groups (dishes). Authors report that motor instability in the negative rumination.
The presentation of the study, technical detail and apparatus are adequate for the required standards, however I would recommend to the authors rewrite of the manuscript to achieve better clarity and improve significance of the this work.
1. I do not understand the premise of the emotional induction (of food preferences) and the experimental task. While it is true that emotions are embodied multi-modally and impact motor system, along with other physiological and hormonal changes, I do not understand the link. Emotions are drivers for actions, in a goal-directed fashion (among many theoretical approaches to emotion you cite works with Frijda and Moors. Are authors suggesting that by 'random' association of negative emotion participants are producing less force in the experimental task. That truly would be a serendipitous finding. What worries me is that there might be a confound between participants drive to continue the task (imagination regarding the unpleasant food) and causing therefore a drop in motivation to perform. What also worries me, is the procedure is so explicit, that participants might want to 'comply' and 'guess' the purpose of the study. There is of course possibility that participants feel disgust, and therefore want to apply less force, as if they would be 'moving' towards this object.
I think in the current version, the underpinning of the discussed and reported findings are vague. I suggest authors reconsider the narrative and try to explain better to the reader the concept they followed designing the study.
2. I understand this is not a re-submission of previous work, but entirely new data sample. If so, the preliminary study can be mentioned somewhere in the introduction, but not in the Abstract, and references throughout the text are not necessary. I understand some methodological concerns were raised with regards to the previous publication, but this is of little relevance to the current assessment.
3. The manuscript is far too detailed and long. It looks like a technical report. And there is even more information in the Supplementary information. I suggest to the authors to simplify and radically cut down the text/content to the most crucial information (for example the single case report is really not necessary, and maybe could be published separately in conference proceedings).
This is a solid study on the technical level. The question of the linkage between emotions and movement, and embodiment of disgust in features such as AF, is fascinating and timely. However, the rationale of the study, along with the in-depth discussion of the affective models that could help to understand the findings. Authors suggest the linkage between the 'mental stress' and 'musculoskeletal complaints'- this is very interesting, but I think pairing the MMT with food imaginations, is not really the path to test this link (it would require perhaps stressing participants in the lab, or measuring their MMT on daily basis and correlating to their daily levels of experiences stress - self-reported/or cardiac activity/skin conductance).
The findings of this study could have application in rehabilitative and clinical context. Therefore I encourage authors to revise this manuscript thoroughly, by shortening the content, clarification of the rationale and incorporating some of the theories of the emotions - be it Fridja, Feldmann-Barrett or LeDoux. At present I am not convinced by the narrative that the 'effect' measured is indeed an impact of the negative emotions/stress on the AF.
I would like to suggest to the authors proofreading by a native speaker as wording seems strange:
'Further investigations remain.'
'underpins the hypothesis'
'related discussion was given'
'is a large agreement'
I apologise if my comments come across harsh, I believe that it is in the best way forward for this work to be revised and resubmitted.
Author Response
Dear Reviewer,
Thank you for your interest, very fast and thorough review and the helpful comments, which were not at all harsh, but on point and comprehensible for us. We tried to address all your concerns in the manuscript and hope we performed the revision to your satisfaction.
You will find the point-by-point-response in the attachement.
Thank you for your effort!
The authors

Reviewer 2 Report
The authors have investigated the influence of pleasant and unpleasant imaginations on the adaptive force in young women. This study is well designed and very interesting. However, there are some difficulties in understanding the methods, indexes of AF, and interpretation of results.
1. Why only women were recruited in the present study? Does gender difference affect the AF?
2. How long did you set the interval between trials? Have you considered the influence of muscle fatigue on the AF?
3. I could not understand how to analyze the AFosc. I think it would be better to explain how to analyze the AFosc with figure.
4. What are the physiological implications of the AFosc? Similarly, please explain the ratio AFosc / AFisomax and the ratio AFosc / AFmax.
5. Can AFmax and AFeccmax be interpreted as equal? Does the difference in length and contraction style of muscles generating the elbow or hip flexion moment affect the AFmax?
6. Did food which participants imagined differ from each trial? If the same, does habituation affect the emotional elicitation?
Author Response
Dear Reviewer,
Thank you for your interest, fast review and the helpful comments.
We hope that we have addressed your concerns to your satisfaction.
You will find the point-by-point-response in the attachement.
Thank you for your effort!
Best regards,
the authors

Round 2
Reviewer 1 Report
Thank you for the extensive reply to my feedback; I really appreciate it and the approach is more clear to me - both in text and in the form of extended explanation.
I am not a native speaker myself, but for the sake of your readership and the impact of your work - please have another exchange with your native speaker colleague, and make sure this is someone familiar with academic style of writing.
P1. Line 33
"A described case example (current stress vs. positive imagery) underpins the hypothesis this approach might support psychomotor diagnostics and therapeutics."
This sentence is not clear - please revise to
'case example is an early evidence that approach presented in the study might have future implications for psychomotor diagnostics and therapeutics'
P2. Line 157
Food imagery was chosen since it is an easy, 'inoffensive'
Please change inoffensive - to 'non-invasive'
P3. Line 158
Delete : 'healthy participants'.
Line 158
Delete: 'The task to enter the emotion is standardized thereby.'
Line 846
Change: 'we presume'
to 'we hypothesize to observe'
Line 851
Change: 'we suppose'
to 'we propose'.
Line 859
Please add a reference for the 'various afferences e.g., nociception'.
I look forward to seeing this in print.
I think this is very valuable contribution to the body of research and the research work itself is of impeccable standard.
Author Respons
Dear Reviewer,
Thank you again for the second round of review. We really appreciate your thorough, precise and helpful comments. It helped to improve the manuscript. And we really appreciate your support for our approach! Thank you very much!
We included all suggestions and the colleague (native English speaker) already read the manuscript again and told us it would be fine. We hope to address your expectations thereby.
Thank you again for the support and your effort!
You will find the point-by-point-response below (and attached).
The authors
Comments Reviewer #1
Thank you for the extensive reply to my feedback; I really appreciate it and the approach is more clear to me - both in text and in the form of extended explanation.
- Thank you for that feedback. We are happy to have addressed your concerns to your satisfaction.
I am not a native speaker myself, but for the sake of your readership and the impact of your work - please have another exchange with your native speaker colleague, and make sure this is someone familiar with academic style of writing.
- Well, at least you have a very good expression in English ?.
Again, we gave the manuscript to the native speaker (already after the first round of review process). He told us it would be fine like that. A few years ago we assigned our manuscripts to an external native speaker and, nevertheless, a reviewer wanted that we revise the English. In the past two years, the reviewers accepted our English style from the beginning. However, we are always grateful to learn better expressions and hope – of course – to meet your requirements. We have included all your suggestions – in case not exactly like you suggested, I explained the reasons (see below)
P1. Line 33
"A described case example (current stress vs. positive imagery) underpins the hypothesis this approach might support psychomotor diagnostics and therapeutics."
This sentence is not clear - please revise to
'case example is an early evidence that approach presented in the study might have future implications for psychomotor diagnostics and therapeutics'
- We changed the sentence to ‘A case example (current stress vs. positive imagery) suggests that approach presented in the study might have future implications for psychomotor diagnostics and therapeutics’
We want to avoid to speak of “evidence” – although we know of the potential and are using it already in therapeutical practice, it seems not to be appropriate to speak of evidence (even not of early evidence) based on one case.
P2. Line 157
Food imagery was chosen since it is an easy, 'inoffensive'
Please change inoffensive - to 'non-invasive'
- We changed it to harmless. ‘Non-invasive’ for us rather refers to medical approaches, although I am aware that ‘invasive’ is also used with respect to the privacy of a person. I hope this is fine with you.
P3. Line 158
Delete : 'healthy participants'.
- done
Line 158
Delete: 'The task to enter the emotion is standardized thereby.'
- done
Line 846
Change: 'we presume'
to 'we hypothesize to observe'
- I really like this wording. Thank you!
Line 851
Change: 'we suppose'
to 'we propose'.
- done
Line 859
Please add a reference for the 'various afferences e.g., nociception'.
- Thank you for that hint, which we consider as very important.
We added two references:- Nijs et al. Nociception affect motor output. J. Pain 2012, 28, 175–181. doi:10.1097/AJP.0b013e318225daf3
- Farina et al. Transient Inhibition of the Human Motor Cortex by Capsaicin-Induced Pain. A Study with Transcranial Magnetic Stimulation. Lett. 2001, 314, 97–101, doi:10.1016/S0304-3940(01)02297-2.
I look forward to seeing this in print.
I think this is very valuable contribution to the body of research and the research work itself is of impeccable standard.
- We really appreciate your thorough, precise and helpful review. It helped to improve the manuscript. And we really appreciate your support for our approach! Thank you very much!
